# MAXIMUM-ENTROPY EXPLORATION WITH FUTURE STATE-ACTION VISITATION MEASURES

## ABSTRACT

Maximum entropy reinforcement learning motivates agents to explore states and actions by providing intrinsic rewards proportional to the entropy of some distribution. In this paper, we study intrinsic rewards proportional to the entropy of the discounted distribution of state-action features visited during future time steps. This approach is motivated by two results. First, we show that this new objective is a lower bound on the standard objective providing intrinsic rewards proportional to the entropy of the discounted distribution of state-action features visited during full trajectories, i.e., starting from initial states. Second, we show that the distribution used in the intrinsic reward definition is the fixed point of a contraction operator. The intrinsic reward can therefore be computed off-policy. We quantify and compare the exploration effectiveness of different maximum entropy objectives. Experiments highlight that the new objective leads to feature exploration concurrent to the alternative methods. In expectation over trajectories, features are typically visited less often, as suggested by the lower bound, but over individual trajectories, features are visited more often than the concurrent approaches. All methods lead to similar control performance on the considered benchmarks.

## 1 INTRODUCTION

Many challenging tasks where an agent makes sequential decisions have been solved with reinforcement learning (RL). Examples range from playing games (Mnih et al., 2015; Silver et al., 2017), or controlling robots (Kalashnikov et al., 2018; Haarnoja et al., 2018a), to managing the energy systems and markets (Boukas et al., 2021; Aittahar et al., 2024). In practice, many RL algorithms are applied in combination with an exploration strategy to achieve high-performance control. Assuming the agent takes actions in a Markov decision process (MDP), these exploration strategies usually consist of providing intrinsic reward bonuses to the agent for achieving certain behaviors. Typically, the bonus enforces taking actions that reduce the uncertainty about the environment (Pathak et al., 2017; Burda et al., 2018; Zhang et al., 2021b), or actions that enhance the variety of states and actions in trajectories (Bellemare et al., 2016; Lee et al., 2019; Guo et al., 2021; Williams & Peng, 1991; Haarnoja et al., 2019). In many of the latter methods, the intrinsic reward function is the entropy of some distribution over the state-action space. Optimizing jointly the reward function of the MDP and the intrinsic reward function, in order to eventually obtain a high-performing policy, is called Maximum Entropy RL (MaxEntRL) and was shown to be effective in many problems.

The reward of the MDP was already extended with the entropy of the policy in early algorithms (Williams & Peng, 1991) and was only later called MaxEntRL (Ziebart et al., 2008; Toussaint, 2009). This particular reward regularization provides substantial improvements in the robustness of the resulting policy (Ziebart, 2010; Husain et al., 2021; Brekelmans et al., 2022) and provides a learning objective function with good smoothness and concavity properties (Ahmed et al., 2019; Bolland et al., 2023). Several commonly used algorithms can be named, like soft Q-learning (Haarnoja et al., 2017; Schulman et al., 2017a) and soft actor-critic (Haarnoja et al., 2018b; 2019). This MaxEntRL objective nevertheless only rewards the randomness of actions and neglects the influences of the policy on the visited states, which, in practice, may lead to inefficient exploration.

In order to enhance exploration, Hazan et al. (2019) were the first to propose to intrinsically motivate agents to have a uniform discounted visitation measure over states. Several works have afterward been developed to maximize the entropy of the discounted state visitation measure and the sta-

tionary state visitation measure. For discrete state and action spaces, optimal exploration policies, which maximize the entropy of these visitation measures, can be computed to near optimality with off-policy tabular model-based RL algorithms (Hazan et al., 2019; Mutti & Restelli, 2020; Tiapkin et al., 2023). For continuous state and action spaces, alternative methods rely on $k$ nearest neighbors to estimate the density of the visitation measure of states (or features built from the states) and compute the intrinsic rewards, which can afterward be optimized with any RL algorithm (Liu & Abbeel, 2021; Yarats et al., 2021; Seo et al., 2021; Mutti et al., 2021). These methods require sampling new trajectories at each iteration; they are on-policy, and estimating the intrinsic reward function is computationally expensive. Some other methods rely on parametric density estimators to reduce the computational complexity and share information across learning steps (Lee et al., 2019; Guo et al., 2021; Islam et al., 2019; Zhang et al., 2021a). The additional function approximator is typically learned on-policy by maximum likelihood estimation based on batches of truncated trajectories. Alternative methods have adapted this MaxEntRL objective to maximize entropy of states visited in single trajectories (Mutti et al., 2022; Jain et al., 2024). When large and/or continuous state and action spaces are involved, relying on parametric function approximators is likely the best choice. Nevertheless, existing algorithms are on-policy. They require sampling new trajectories from the environment at (nearly) every update of the policy, and cannot be applied using a buffer of arbitrary transitions, in batch-mode RL, or in continuing tasks. Furthermore, learning the discounted visitation measure is more desirable than learning the stationary one, but may be challenging due to the exponentially decreasing influence of the time step at which states are visited (Islam et al., 2019).

The main contribution of this paper is to introduce a MaxEntRL objective relying on a new intrinsic reward function for exploring effectively the state and action spaces, which also alleviates the previous limitations. This intrinsic reward function is the relative entropy of the discounted distribution of state-action features visited during future time steps. We prove two results motivating the MaxEntRL objective. First, this new objective is a lower bound on the MaxEntRL objective previously described, which integrates the marginal visitation distribution of states and actions. Second, the visitation distribution used in the new intrinsic reward function is the fixed point of a contraction operator. The intrinsic reward can therefore efficiently be computed off-policy using N-step state-action transitions and bootstrapping the operator. It is then possible to approximate the intrinsic reward function and learn a policy maximizing the extended rewards with existing RL algorithms. We demonstrate in our experiments how to adapt soft actor-critic (Haarnoja et al., 2018b) to optimize the new objective, and we compare the effectiveness of exploration with different maximum entropy objectives. In expectation over trajectories, features are typically visited less often, as suggested by the lower bound, but over individual trajectories, features are visited more often than the concurrent approaches. All methods lead to similar control performance on the considered benchmarks.

The visitation measure of future states and actions, which we use to extend the reward function in this article, has a well-established history in the development of RL algorithms. It was popularized by Janner et al. (2020), who learned the distribution of future states as a generalization of the successor features (Barreto et al., 2017). They demonstrated that this distribution allows expressing the state-action value function by separating the influence of the dynamics and the reward function, and that it could be learned off-policy by exploiting its recursive expression. Several algorithms have been proposed to learn this distribution, either by maximum likelihood estimation (Janner et al., 2020), by contrastive learning (Mazoure et al., 2023b), or using diffusion models (Mazoure et al., 2023c). These distributions of future states and actions have found applications in goal-based RL (Eysenbach et al., 2020; 2022), in offline pre-training with expert examples (Mazoure et al., 2023a), in model-based RL (Ma et al., 2023), or in planning (Eysenbach et al., 2023). We are the first to integrate them into the MaxEntRL framework for enhancing exploration. Concurrent to our work Mohamed et al. developed a similar intrinsic reward dependent on the distribution of future states.

The manuscript is organized as follows. In Section 2, the RL problem and the MaxEntRL framework are formulated. In Section 3, we introduce and discuss a new MaxEntRL objective. Section 4 details how to learn a model of the conditional state visitation probability that allows estimating this new objective. We finally present experimental results in Section 5 and conclude in Section 6.

## 2 BACKGROUND AND PRELIMINARIES

### 2.1 MARKOV DECISION PROCESSES

This paper focuses on problems in which an agent makes sequential decisions in a stochastic environment (Sutton & Barto, 2018). The environment is modeled with an infinite-time Markov decision process (MDP) composed of a state space $\mathcal{S}$, an action space $\mathcal{A}$, an initial state distribution $p_0$, a transition distribution $p$, a bounded reward function $R$, and a discount factor $\gamma \in [0, 1)$. Agents interact in this MDP by providing actions sampled from a policy $\pi$. During this interaction, an initial state $s_0 \sim p_0(\cdot)$ is first sampled, then, the agent provides at each time step $t$ an action $a_t \sim \pi(\cdot|s_t)$ leading to a new state $s_{t+1} \sim p(\cdot|s_t, a_t)$. In addition, after each action $a_t$ is executed, a reward $r_t = R(s_t, a_t) \in \mathbb{R}$ is observed. We denote the expected return of the policy $\pi$ by

$$J(\pi) = \mathop{\mathbb{E}}_{\substack{s_0 \sim p_0(\cdot) \\ a_t \sim \pi(\cdot|s_t) \\ s_{t+1} \sim p(\cdot|s_t, a_t)}} \left[ \sum_{t=0}^{\infty} \gamma^t R(s_t, a_t) \right] . \tag{1}$$

An optimal policy $\pi^*$ is one with maximum expected return

$$\pi^* \in \arg\max_{\pi} J(\pi) . \tag{2}$$

### 2.2 MAXIMUM ENTROPY REINFORCEMENT LEARNING

In maximum entropy reinforcement learning (MaxEntRL) an optimal policy $\pi^*$ is approximated by maximizing a surrogate objective function $L(\pi)$, where the reward function from the MDP is extended by an intrinsic reward function. The latter is the (relative) entropy of some particular distribution. A general definition of the MaxEntRL objective function is

$$L(\pi) = \mathop{\mathbb{E}}_{\substack{s_0 \sim p_0(\cdot) \\ a_t \sim \pi(\cdot|s_t) \\ s_{t+1} \sim p(\cdot|s_t, a_t)}} \left[ \sum_{t=0}^{\infty} \gamma^t \left( R(s_t, a_t) + \lambda R^{int}(s_t, a_t) \right) \right] , \tag{3}$$

where this objective depends on the intrinsic reward function $R^{int}$. We propose a generic formulation that, to the best of our knowledge, encompasses most existing intrinsic rewards from the literature. Given a feature space $\mathcal{Z}$, a conditional feature distribution $q^\pi : \mathcal{S} \times \mathcal{A} \to \Delta(\mathcal{Z})$, depending on the policy $\pi$, and a relative measure $q^* \in \Delta(\mathcal{Z})$, the MaxEntRL intrinsic reward function is

$$R^{int}(s, a) = -KL_z \left[ q^\pi(z|s, a) \| q^*(z) \right] = \mathop{\mathbb{E}}_{z \sim q^\pi(\cdot|s, a)} \left[ \log q^*(z) - \log q^\pi(z|s, a) \right] . \tag{4}$$

Importantly, the intrinsic reward function is (implicitly) dependent on the policy $\pi$ through the distribution $q^\pi$. We define an optimal exploration policy as a policy that maximizes the expected sum of discounted intrinsic rewards only. Note that a policy maximizing $L(\pi)$ is generally not optimal, due to the potential gap between the optimum of the return $J(\pi)$ and the optimum of the learning objective $L(\pi)$. This subject is inherent to exploration with intrinsic rewards (Bolland et al., 2024).

MaxEntRL algorithms optimize objective functions as defined in equation equation 3 depending on some intrinsic reward function that can be expressed as in equation equation 4. The particularity of each algorithm is its estimation of the intrinsic reward and of the stochastic gradient of the learning objective. Often, a pseudo reward $\log q^*(z) - \log q^\pi(z|s, a)$ is computed from a sample $z \sim q^\pi(\cdot|s, a)$ to extend the MDP reward function and used by an existing RL algorithm.

Many of the existing MaxEntRL algorithms optimize an objective that depends on the entropy of the policy for exploring the action space (Haarnoja et al., 2018b; Toussaint, 2009). The feature space is then the actions space $\mathcal{Z} = \mathcal{A}$, and the conditional feature distribution is the policy $q^\pi(z|s, a) = \pi(z|s)$, for all $a$. Other algorithms optimize objectives enhancing state space exploration (Hazan et al., 2019; Lee et al., 2019; Islam et al., 2019; Guo et al., 2021). The feature space is the state space $\mathcal{Z} = \mathcal{S}$. The conditional feature distribution $q^\pi(z|s, a)$ is either the marginal probability of states in trajectories of $T$ time steps, or the discounted state visitation measure, for all $s$ and $a$. In the literature, the relative measure $q^*(z)$ is usually a uniform distribution, and the relative entropy is computed as the differential entropy, i.e., by neglecting $\log q^*(z)$ in equation equation 4. In continuous spaces, the latter is ill-defined and other relative measures may be used.

## 3 MaxEntRL with Visitation Distributions

### 3.1 Definition of the MaxEntRL Objective

In the following, we introduce a new MaxEntRL intrinsic reward based on the conditional state-action visitation probability $d^{\pi,\gamma}(\bar{s},\bar{a}|s,a)$ and the conditional state visitation probability $d^{\pi,\gamma}(\bar{s}|s,a)$

$$d^{\pi,\gamma}(\bar{s},\bar{a}|s,a) = (1-\gamma)\pi(\bar{a}|\bar{s})\sum_{\Delta=1}^{\infty}\gamma^{\Delta-1}p_{\Delta}^{\pi}(\bar{s}|s,a) \tag{5}$$

$$d^{\pi,\gamma}(\bar{s}|s,a) = (1-\gamma)\sum_{\Delta=1}^{\infty}\gamma^{\Delta-1}p_{\Delta}^{\pi}(\bar{s}|s,a)\,, \tag{6}$$

where $p_{\Delta}^{\pi}$ is the transition probability in $\Delta$ time steps with the policy $\pi$. The distribution from equation equation 5 can be factorized as a function of the distribution from equation equation 6 such that $d^{\pi,\gamma}(\bar{s},\bar{a}|s,a) = \pi(\bar{a}|\bar{s})d^{\pi,\gamma}(\bar{s}|s,a)$. The conditional state (respectively, state-action) visitation probability distribution measures the states (respectively, states and actions) that are visited on expectation over infinite trajectories starting from a state and an action. Both distributions generalize the (marginal discounted) state visitation probability measure (Manne, 1960).

**Definition 3.1.** Let us consider the feature space $\mathcal{Z}$ and a feature distribution $h : \mathcal{S} \times \mathcal{A} \to \Delta(\mathcal{Z})$. The intrinsic reward is defined by equation equation 4, for any relative measure $q^*$, with conditional distribution

$$q^{\pi}(z|s,a) = \int h(z|\bar{s},\bar{a})d^{\pi,\gamma}(\bar{s},\bar{a}|s,a)\,d\bar{s}\,d\bar{a}\,. \tag{7}$$

Optimal exploration policies are here intrinsically motivated to take actions so that the discounted visitation measure of future features is distributed according to $q^*$ in each state and for each action. It allows to select features that must be visited during trajectories according to prior knowledge about the problem if any. Alternatively, it allows to only explore lower dimensional feature spaces, or to explore sufficient statistics from the state-action pairs.

Existing RL algorithms can be used to compute policies according to the MaxEntRL definition 3.1 computing for each state $s$ and action $a$ the additional (pseudo) reward

$$R^{int}(s,a) = \log q^*(z) - \log q^{\pi}(z|s,a)\,, \tag{8}$$

where $z \sim q^{\pi}(\cdot|s,a)$. This reward is a single-sample Monte-Carlo estimate of equation equation 4, unbiased for fixed $q^{\pi}$. This computation requires sampling features $z$ from the conditional distribution $q^{\pi}$ and estimating the probability of these samples $q^{\pi}(z|s,a)$. It can be achieved by solving the integral equation equation 7, e.g., numerically by sampling states-actions $(\bar{s},\bar{a}) \sim d^{\pi,\gamma}(\cdot,\cdot|s,a)$ and features $z \sim h(\cdot|\bar{s},\bar{a})$, or learning a model of the conditional feature distribution. Section 4 provides a method for learning such a model off-policy.

### 3.2 Relationship with Alternative MaxEntRL Objectives

Let us now consider for simplicity that the feature distribution is the identity mapping, so that $z = (\bar{s},\bar{a})$. According to Definition 3.1, we then have $q^{\pi}(z|s,a) = d^{\pi,\gamma}(\bar{s},\bar{a}|s,a) = d^{\pi,\gamma}(\bar{s}|s,a)\pi(\bar{a}|\bar{s})$. This MaxEntRL intrinsic reward function can be compared to one where the conditional distribution is $q^{\pi}(z|s,a) = d^{\pi,\gamma}(\bar{s},\bar{a}) = d^{\pi,\gamma}(\bar{s})\pi(\bar{a}|\bar{s})$, for all $s$ and $a$, and where the MaxEntRL objective function without MDP rewards simplifies to the negated entropy of that distribution. Theorem 3.2, shown in Appendix A, relates the two objectives with no MDP rewards.

**Theorem 3.2.** *For any policy $\pi$ and any relative measure $q^*$, the marginal and conditional visitation measures satisfy*

$$\mathbb{E}_{s,a\sim d^{\pi,\gamma}(\cdot,\cdot)}\left[-KL_{\bar{s},\bar{a}}\left[d^{\pi,\gamma}(\bar{s},\bar{a}|s,a)||q^*(\bar{s},\bar{a})\right]\right]$$

$$\leq -KL_{\bar{s},\bar{a}}(d^{\pi,\gamma}(\bar{s},\bar{a})||q^*(\bar{s},\bar{a})) + L\sqrt{2\,KL_{\bar{s},\bar{a}}(d^{\pi,\gamma}(\bar{s},\bar{a})||\tilde{d}^{\pi,\gamma}(\bar{s},\bar{a}))}\,,$$

*where $L$ is a constant and $\tilde{d}^{\pi,\gamma}(\bar{s},\bar{a}) = \mathbb{E}_{s,a\sim d^{\pi,\gamma}(\cdot,\cdot)}[d^{\pi,\gamma}(\bar{s},\bar{a}|s,a)]$.*

Let us first assume that $\tilde{d}^{\pi,\gamma}(\bar{s},\bar{a}) = d^{\pi,\gamma}(\bar{s},\bar{a})$. Then, the left-hand side corresponds to our new objective with $R(s,a) = 0$, which is a lower bound on the negated entropy of the marginal state-action visitation measure. If $d^{\pi,\gamma}(\bar{s},\bar{a}|s,a) = q^*(\bar{s},\bar{a})$ almost everywhere, then the lower bound is zero, and $d^{\pi,\gamma}(\bar{s},\bar{a}) = q^*(\bar{s},\bar{a})$ almost everywhere as well. It implies that our MaxEntRL objective is a lower bound on the alternative MaxEntRL objective with marginal visitation. Additionally, a policy maximizing our objective also maximizes the alternative objective, when achieving an intrinsic reward of zero. In the limit when the discount factor tends to one $\gamma \to 1$, this equivalence also holds as the effect of the initial state vanishes and both distributions converge to the stationary visitation distribution, provided it exists. In practice, the bound holds only when $\tilde{d}^{\pi,\gamma}(\bar{s},\bar{a})$ is close to $d^{\pi,\gamma}(\bar{s},\bar{a})$; it intuitively corresponds to a stationary assumption on the initial states.

This result connects MaxEntRL optimizing the entropy of the conditional visitation with MaxEntRL optimizing the entropy of the same distribution marginalized over initial states and actions. It can be straightforwardly extended to relate the conditional and marginal visitation of features, typically such that MaxEntRL algorithms optimizing intrinsic rewards from Definition 3.1 with $z = \bar{s}$ maximize lower bounds of MaxEntRL algorithms using the state visitation (Hazan et al., 2019).

# 4 OFF-POLICY LEARNING OF CONDITIONAL VISITATION MODELS

## 4.1 FIXED-POINT PROPERTIES OF CONDITIONAL VISITATION

As explained in Section 3.1, the MaxEntRL intrinsic reward function in Definition 3.1 can be computed sampling from the conditional feature distribution and evaluating the probability of these samples. In this section, we establish useful properties of this conditional distribution.

Let us first recall that the conditional state-action visitation distribution accepts a recursive definition (Janner et al., 2020) that is a trivial fixed point of the operator $\mathcal{T}^\pi$ from Definition 4.1.

**Definition 4.1.** The operator $\mathcal{T}^\pi$ is defined over the space of conditional state-action distribution as
$$\mathcal{T}^\pi q(\bar{s},\bar{a}|s,a) = (1-\gamma)\pi(\bar{a}|\bar{s})p(\bar{s}|s,a) + \gamma \mathop{\mathbb{E}}_{\substack{s' \sim p(\cdot|s,a) \\ a' \sim \pi(\cdot|s')}} [q(\bar{s},\bar{a}|s',a')] \,.$$

Theorem 4.2 establishes that the operator $\mathcal{T}^\pi$ is a contraction mapping, which furthermore implies the uniqueness of its fixed point. Assuming the result of the operator could be computed (or estimated), the fixed point could also be computed by successive application of this operator. It would allow to compute the conditional state-action visitation distribution, and the intrinsic reward function.

**Theorem 4.2.** *The operator $\mathcal{T}^\pi$ is $\gamma$-contractive in $\bar{L}_n$-norm, where $\bar{L}_n(f)^n = \sup_y \int |f(x|y)|^n dx$.*

Definition 4.3 introduces the operator $\mathcal{P}^\pi$ that implicitly depends on the feature distribution $h$. If this distribution is the identity map, then both operators $\mathcal{P}^\pi$ and $\mathcal{T}^\pi$ are equal.

**Definition 4.3.** The operator $\mathcal{P}^\pi$ is defined over the space of conditional feature distribution as
$$\mathcal{P}^\pi q(z|s,a) = \mathop{\mathbb{E}}_{\substack{s' \sim p(\cdot|s,a) \\ a' \sim \pi(\cdot|s')}} [(1-\gamma)h(z|s',a') + \gamma q(z|s',a')] \,.$$

Following, we establish in Theorem 4.4 that this operator is also a contraction mapping, such that Theorem 4.2 can be considered a corollary of the present result. The fixed point could again theoretically be computed by successive application of this operator.

**Theorem 4.4.** *The operator $\mathcal{P}^\pi$ is $\gamma$-contractive in $\bar{L}_n$-norm, where $\bar{L}_n(f)^n = \sup_y \int |f(x|y)|^n dx$.*

Finally, we establish in Theorem 4.5 that the unique fixed point of $\mathcal{P}^\pi$ is the conditional distribution used in the MaxEntRL intrinsic reward from Definition 3.1. Assuming we could approximate this fixed point, we would get a model to compute the reward as in equation equation 8.

**Theorem 4.5.** *The unique fixed point of the operator $\mathcal{P}^\pi$ is*
$$q^\pi(z|s,a) = \int h(z|\bar{s},\bar{a})d^{\pi,\gamma}(\bar{s},\bar{a}|s,a) \, d\bar{s}\, d\bar{a} \,.$$

The theorems are shown in Appendix B.

## 4.2 TD Learning of Conditional Visitation Models

In practice, computing the result of the operator $\mathcal{P}^\pi$ (and $(\mathcal{P}^\pi)^N$ after $N$ applications) may be intractable when large state and action spaces are at hand or when these spaces are continuous. It furthermore requires having a model of the MDP. A common approach is then to rely on a function approximator $q_\psi$ to approximate the fixed point. Furthermore, similarly to TD-learning methods (Sutton & Barto, 2018), Theorem 4.4 suggests to optimize the parameters of this model $q_\psi$ to minimize the residual of the operator, measured with an $\bar{L}_n$-norm for which the operator is $\gamma$-contractive. With this metric, estimating the residual would require estimating the probability of future features, and cannot be trivially minimized by stochastic gradient descent using transitions from the environment. We therefore propose to solve a surrogate minimum cross-entropy problem, in which stochastic gradient descent can be applied afterwards. For any policy $\pi$, the fixed point $q^\pi$ is approximated with a function approximator $q_\psi$ with parameter $\psi$ optimized to solve

$$\arg\min_{\psi} \mathop{\mathbb{E}}_{\substack{s,a \sim g(\cdot,\cdot) \\ \bar{z} \sim (\mathcal{P}^\pi)^N q_\psi(\cdot|s,a)}} \left[ -\log q_\psi(\bar{z}|s,a) \right] , \tag{9}$$

where $g$ is an arbitrary distribution over the state and action space, and where $N$ is any positive integer. This optimization problem is related to minimizing the KL-divergence instead of an $\bar{L}_n$-norm (Bishop & Nasrabadi, 2006).

Let us make explicit how samples from the distribution $(\mathcal{P}^\pi)^N q_\psi(\bar{z}|s,a)$ can be generated from the MDP. By definition of the operator $\mathcal{P}^\pi$, the distribution $(\mathcal{P}^\pi)^N q_\psi(\bar{z}|s,a)$ is the mixture

$$(\mathcal{P}^\pi)^N q_\psi(\bar{z}|s,a) = \left( \sum_{\Delta=1}^{N} (1-\gamma)\gamma^{\Delta-1} \mathop{\mathbb{E}}_{\substack{s' \sim p_\Delta^\pi(\cdot|s,a) \\ a' \sim \pi(\cdot|s')}} [h(\bar{z}|s',a')] \right) + \gamma^N \mathop{\mathbb{E}}_{\substack{s' \sim p_N^\pi(\cdot|s,a) \\ a' \sim \pi(\cdot|s')}} [q_\psi(\bar{z}|s',a')]$$

$$= \sum_{\Delta=1}^{\infty} \mathcal{G}_{1-\gamma}(\Delta) \mathop{\mathbb{E}}_{\substack{s' \sim p_{\Delta'}^\pi(\cdot|s,a) \\ a' \sim \pi(\cdot|s')}} [b_\psi(\bar{z}|s',a',\Delta)] \Bigg|_{\Delta'=\min(\Delta,N)} , \tag{10}$$

where $\mathcal{G}_{1-\gamma}(\Delta)$ is the probability of $\Delta$ from a geometric distribution of parameter $1-\gamma$, and

$$b_\psi(\bar{z}|s,a,\Delta) = \left\{ \begin{array}{ll} h(\bar{z}|s,a) & \Delta \leq N \\ q_\psi(\bar{z}|s,a) & \Delta > N \end{array} \right. \tag{11}$$

Sampling from $(\mathcal{P}^\pi)^N q_\psi(\bar{s}|s,a)$ consists in sampling from the mixture.

Let us reformulate the problem equation equation 9 highlighting the elements from the mixture and applying importance weighting

$$\arg\min_{\psi} \mathop{\mathbb{E}}_{\substack{s,a \sim g(\cdot,\cdot) \\ \Delta \sim \mathcal{G}_{1-\gamma'}(\cdot) \\ s' \sim p_{\Delta'}^\beta(\cdot|s,a) \\ a' \sim \pi(\cdot|s') \\ \bar{z} \sim b_\psi(\cdot|s',a',\Delta)}} \left[ -\frac{\mathcal{G}_{1-\gamma}(\Delta)}{\mathcal{G}_{1-\gamma'}(\Delta)} \frac{p_{\Delta'}^\pi(s'|s,a)}{p_{\Delta'}^\beta(s'|s,a)} \log q_\psi(\bar{z}|s',a') \right] \Bigg|_{\Delta'=\min(\Delta,N)} . \tag{12}$$

Importance weighting is applied to the transition probability $p_{\Delta'}^\pi$ and to the geometric distribution $\mathcal{G}_{1-\gamma}$. It introduces the behavior policy $\beta$ and a pseudo discount factor $\gamma'$. The first allows off-policy learning, and the second helps avoid that some elements of the mixture have negligible probabilities, improving results in practice. When $\beta = \pi$ or when $N = 1$, the second importance ratio simplifies to one; otherwise, it simplifies to a (finite) product of ratios of policies.

Learning $q_\psi$ from samples can be achieved by solving problem equation equation 12 as an intermediate step to any RL algorithm. First, the objective function is estimated using transitions. Second, this estimate is differentiated, and the parameter $\psi$ is updated by gradient descent steps. In practice, the gradients generated by differentiating this loss function are biased. The influence of the parameter $\psi$ on the probability of the sample $z$ is neglected when bootstrapping, i.e., the partial derivative of $(\mathcal{P}^\pi)^N q_\psi(\bar{s}|s_t,a_t)$ with respect to $\psi$ is neglected, and a target network is used. This is analogous to SARSA and TD-learning strategies (Sutton & Barto, 2018). Furthermore, we suggest neglecting the ratio of transition probabilities, which introduces a dependency of the distribution $q_\psi$ on the policy $\beta$. Finally, the model $q_\psi$ is used to compute the intrinsic rewards and update the policy.

## 5 EXPERIMENTS

### 5.1 EXPERIMENTAL SETTING

In this section, we detail the methodology applied to compare the MaxEntRL objectives discussed above. We use a suite of adapted environments for illustration, adapt algorithms from the literature to isolate the effect of the choice of objective from algorithmic considerations, and introduce measures to quantify the quality of exploration.

**Environments.** Experiments are performed on environments from the Minigrid suite (Chevalier-Boisvert et al., 2023). In the latter, an agent must travel across a grid containing walls and passages in order to reach a goal. The state space is a full observation of the maze, represented with an image, and the agent's orientation. The agent can take four different actions: turn left, turn right, move forward, or stand still. The need for exploration comes from the sparsity of the reward function, which is zero everywhere and equals one in the state to be reached.

**Exploration strategies.** We compare three exploration strategies, i.e., three intrinsic reward functions and their corresponding algorithm. The first exploration strategy motivates agents to visit actions uniformly. The feature space is the action space $\mathcal{Z} = \mathcal{A}$, the conditional feature distribution is the policy $q^\pi(z|s,a) = \pi(z|s)$ for all $a$, and the relative measure $q^*$ is uniform. The second exploration strategy motivates agents to have uniform (marginal) discounted visitation measures as originally proposed by Hazan et al. (2019) and discussed in Section 1. Here, the features $z \in \mathcal{Z}$ are the positions of the agent in the environment, the conditional distribution $q^\pi(z|s,a)$ is the discounted visitation measure of features for each state $s$ and action $a$, and the relative measure $q^*$ is uniform. The last exploration strategy is the one presented in Section 3. Again, the features $z \in \mathcal{Z}$ are the positions of the agent in the environment, and the relative measure $q^*$ is uniform.

**Algorithms.** Existing algorithms can be adapted to optimize the previous MaxEntRL objective by adding the intrinsic reward to the reward from the MDP during policy optimization. In some algorithms, as in ours, it requires learning an additional model of some visitation measure. In this paper we adapted soft actor-critic to incorporate additional intrinsic reward functions. We consider three variants, one for each exploration strategy. First, without additional intrinsic reward, it is already a MaxEntRL algorithm that enforces the entropy of the policy. Second, we adapt the algorithm from Zhang et al. (2021a), using SAC (Degris et al., 2012) instead of PPO (Schulman et al., 2017b) to improve sample efficiency and using a categorical distribution rather than a variational auto-encoder to approximate the visitation measure, which is made possible as the state-action space is discrete. It allows optimizing the approximator without relying on the evidence lower bound. Third, we adapt SAC to incorporate our reward function as discussed previously and detailed in Appendix C.

**Exploration metrics.** The last step is to quantify the quality of the exploration policies. Most often, it consists in measuring the diversity of states (or features) visited when only optimizing the intrinsic rewards, i.e., when $R(s,a) = 0$. Here we report two such metrics during learning. First, the entropy of features visited by policies

$$-KL_z(d^{\pi,\gamma}(z)||q^*(z)),\qquad(13)$$

where $d^{\pi,\gamma}(z) = \mathbb{E}_{s,a\sim d^{\pi,\gamma}(\cdot,\cdot)}[h(z|s,a)] = q^\pi(z|s,a)$ for all $a$ and $s$. Second, the conditional entropy of features visited by policies given the initial state, on expectation,

$$-\mathbb{E}_{s_0\sim p_0(\cdot)}\left[KL_z(d^{\pi,\gamma}(z|s_0)||q^*(z))\right],\qquad(14)$$

where $d^{\pi,\gamma}(z|s_0) = \mathbb{E}_{s,a\sim d^{\pi,\gamma}(\cdot,\cdot|s_0)}[h(z|s,a)]$. Both metrics measure the diversity of features visited. The first measures the expectation over episodes, whereas the second metric focuses on the diversity within individual episodes. The first metric also corresponds to the exploration objectives maximized by the second algorithm.

### 5.2 EXPERIMENTAL RESULTS

The methodology explained above is applied to various environments, for which the evolution of metrics is reported in Appendix D. Details on the hyperparameters are given in Appendix E. We summarize the observations in this section.

**Marginal exploration.** Figure 1 in Appendix D illustrates the evolution of the entropy of features visited by policies $-KL_z(d^{\pi,\gamma}(z)||q^*(z))$ as a function of the learning iterations, when only optimizing the intrinsic rewards. We distinguish two situations. First, for some environments, the entropy does not evolve much during learning, and the three exploration strategies perform similarly. This is mostly due to the influence of the initial state distribution. In several environments, the initial position is drawn uniformly at random such that the entropy of a random policy leads to high feature entropy due to symmetries. Finding optimal policies in such environments is arguably easy, as a large diversity of transitions will be observed without requiring complex MaxEntRL objectives. Second, for other environments, the entropy increases rapidly for the second algorithm, with marginal visitation measures (MV), and for the third algorithm, with conditional visitation measures (CV), and a high-entropy policy results from the optimization. In these environments, MV achieves the highest entropy, followed closely by CV, while SAC performs poorly. It is worth noting that CV challenges the concurrent method despite optimizing a different objective and even outperforms the benchmark in some situations. Here the environments are apparently more complex to explore, and both advanced strategies allow observing a wide diversity of features on expectation.

**Episode exploration.** Figure 2 in Appendix D illustrates the evolution of the second metric $-\mathbb{E}_{s_0 \sim p_0(\cdot)}\left[KL_z(d^{\pi,\gamma}(z|s_0)||q^*(z))\right]$ as a function of the learning iterations, when only optimizing the intrinsic rewards. In opposition to the previous case, maximum-entropy exploration improves the entropy as a function of the learning iterations in most environments. Even in environments with uniform initial positions, motivating agents to explore leads to a larger variety of features visited during individual trajectories. Again, there are nevertheless situations where the three exploration strategies lead to similar entropy. For most environments, the intrinsic reward we present in this paper performs better than alternative exploration strategies.

**Control policies.** The literature sometimes focuses on exploration only, but MaxEntRL usually aims to explore in order to eventually compute a high-performance policy. Figure 3 in Appendix D reports the evolution of the expected return of policies when combining rewards from the environment with intrinsic rewards. In most environments, a sufficiently large weight to environment rewards combined with a large buffer leads to improving return during learning. In some environments, the complex exploration strategies allow learning faster, but no significant improvement was observed for the most favorable hyperparameters.

## 5.3 Discussion of Experiments

Experiments highlight that complex MaxEntRL methods are sometimes necessary to achieve better feature space coverage. In particular, our method allows better exploration of features within individual trajectories compared to more standard objectives. A likely justification is that our objective, relying on the conditional entropy, motivates agents to explore in the future, i.e., for the remainder of the trajectory, where alternative objectives are more influenced by the initial actions. To the best of our knowledge, it is unclear if there is a best metric to compare exploration strategies, in particular when experiments seem to highlight that exploration is highly environment dependent.

Several phenomena influence the learning of the visitation models. First, when $\gamma$ is close to one, the learning becomes unstable in practice. We hypothesize it results from the increased importance of future states. Increasing parameter $N$ helps mitigate the issue as there is less bootstrapping, reducing the risk of learning a biased target. Second, we neglect some importance weights in practice to reduce variance, which makes $q_\psi$ partially dependent on the behavior policy $\beta$. Bootstrapping still propagates long-term effects of the policy. In practice, providing policy regularization when using the two complex exploration objectives also appeared to be critical to stabilize learning.

Finally, we relied on off-policy actor-critic for concreteness, yet the MaxEntRL objective is agnostic to the control backbone, and similar results should hold with other RL methods. We decided to share the same backbone to center the discussion around the MaxEntRL objective but acknowledge that the choice of algorithm may influence practical performance. Also, we observe our method offers a practical alternative to directly maximizing marginal visitation, but we did not focus on potential theoretical advantages of different exploration objectives.

## 6 CONCLUSION

In this paper, we presented a new MaxEntRL objective providing intrinsic reward bonuses proportional to the entropy of the distribution of features built from the states and actions visited by the agent in future time steps. The reward bonus can be estimated efficiently by sampling from the conditional distribution of features visited, which we proved to be the fixed point of a contraction mapping and can be learned for any policy relying on batches of arbitrary transitions. We propose an end-to-end off-policy algorithm maximizing our objective that allows exploring effectively the state and action spaces. The algorithm is benchmarked on several control problems. The method we developed is easy to implement and can be integrated into already existing RL algorithms.

Future work should focus on benchmarking the method in more challenging environments, including environments with larger or continuous state-action spaces. For the continuous case, this will require adapting the density estimator and the algorithm accordingly. Finally, in this paper, the feature space to explore is fixed a priori, but could be learned. A potential avenue is to explore reward-predictive feature spaces.

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

# A PROOF LOWER BOUND

**Proof Theorem 3.2** Let $\tilde{d}^{\pi,\gamma}(\bar{s},\bar{a}) = \mathbb{E}_{s,a \sim d^{\pi,\gamma}(\cdot,\cdot)}[d^{\pi,\gamma}(\bar{s},\bar{a}|s,a)]$. Let us develop using the convexity of KL divergence, positiveness of KL divergence, integral properties and Pinsker's inequality

$$\mathbb{E}_{s,a \sim d^{\pi,\gamma}(\cdot,\cdot)}\left[KL_{\bar{s},\bar{a}}\left[d^{\pi,\gamma}(\bar{s},\bar{a}|s,a)||q^*(\bar{s},\bar{a})\right]\right]$$

$$= \mathbb{E}_{\substack{s,a \sim d^{\pi,\gamma}(\cdot,\cdot) \\ \bar{s},\bar{a} \sim d^{\pi,\gamma}(\cdot,\cdot|s,a)}}\left[\log \frac{d^{\pi,\gamma}(\bar{s},\bar{a}|s,a)}{q^*(\bar{s},\bar{a})}\right]$$

$$= \mathbb{E}_{\substack{s,a \sim d^{\pi,\gamma}(\cdot,\cdot) \\ \bar{s},\bar{a} \sim d^{\pi,\gamma}(\cdot,\cdot|s,a)}}\left[\log \frac{d^{\pi,\gamma}(\bar{s},\bar{a}|s,a)}{d^{\pi,\gamma}(\bar{s},\bar{a})} + \log \frac{d^{\pi,\gamma}(\bar{s},\bar{a})}{q^*(\bar{s},\bar{a})}\right]$$

$$\geq \mathbb{E}_{\substack{s,a \sim d^{\pi,\gamma}(\cdot,\cdot) \\ \bar{s},\bar{a} \sim d^{\pi,\gamma}(\cdot,\cdot|s,a)}}\left[\log \frac{d^{\pi,\gamma}(\bar{s},\bar{a})}{q^*(\bar{s},\bar{a})}\right]$$

$$= \int d^{\pi,\gamma} \log \frac{d^{\pi,\gamma}}{q^*} - \int (d^{\pi,\gamma} - \tilde{d}^{\pi,\gamma}) \log \frac{d^{\pi,\gamma}}{q^*}$$

$$= KL_{\bar{s},\bar{a}}(d^{\pi,\gamma}(\bar{s},\bar{a})||q^*(\bar{s},\bar{a})) - \int (d^{\pi,\gamma} - \tilde{d}^{\pi,\gamma}) \log \frac{d^{\pi,\gamma}}{q^*}$$

$$\geq KL_{\bar{s},\bar{a}}(d^{\pi,\gamma}(\bar{s},\bar{a})||q^*(\bar{s},\bar{a})) - \left|\int (d^{\pi,\gamma} - \tilde{d}^{\pi,\gamma}) \log \frac{d^{\pi,\gamma}}{q^*}\right|$$

$$\geq KL_{\bar{s},\bar{a}}(d^{\pi,\gamma}(\bar{s},\bar{a})||q^*(\bar{s},\bar{a})) - L \int \left|d^{\pi,\gamma} - \tilde{d}^{\pi,\gamma}\right|$$

$$\geq KL_{\bar{s},\bar{a}}(d^{\pi,\gamma}(\bar{s},\bar{a})||q^*(\bar{s},\bar{a})) - 2\,L\,TV(d^{\pi,\gamma}, \tilde{d}^{\pi,\gamma})$$

$$\geq KL_{\bar{s},\bar{a}}(d^{\pi,\gamma}(\bar{s},\bar{a})||q^*(\bar{s},\bar{a})) - L\sqrt{2\,KL_{\bar{s},\bar{a}}(d^{\pi,\gamma}(\bar{s},\bar{a})||\tilde{d}^{\pi,\gamma}(\bar{s},\bar{a}))}$$

## B  PROOFS THEOREMS ON CONTRACTIONS

**Proof Theorem 4.2.**  This theorem is a particular case of Theorem 4.4, where the feature space $\mathcal{Z} = \mathcal{S} \times \mathcal{A}$ and the mapping $h$ a Dirac

$$h(z|s,a) = \delta_{(s,a)}(z) \,.$$

$\square$

**Proof Theorem 4.4.**  For all conditional distributions $p$ and $q$

$$\sup_{s,a} L_n(\mathcal{P}^\pi p(\cdot|s,a), \mathcal{P}^\pi q(\cdot|s,a))^n = \sup_{s,a} \int |\mathcal{P}^\pi p(\bar{z}|s,a) - \mathcal{P}^\pi q(\bar{z}|s,a)|^n \, d\bar{z}$$

$$= \gamma \sup_{s,a} \int \left| \mathop{\mathbb{E}}_{\substack{s' \sim p_1(\cdot|s,a) \\ a' \sim \pi(\cdot|s')}} [p(\bar{z}|s',a') - q(\bar{z}|s',a')] \right|^n \, d\bar{z}$$

$$\leq \gamma \sup_{s,a} \int \mathop{\mathbb{E}}_{\substack{s' \sim p_1(\cdot|s,a) \\ a' \sim \pi(\cdot|s')}} [|p(\bar{z}|s',a') - q(\bar{z}|s',a')|^n] \, d\bar{z}$$

$$= \gamma \sup_{s,a} \mathop{\mathbb{E}}_{\substack{s' \sim p_1(\cdot|s,a) \\ a' \sim \pi(\cdot|s')}} \left[ \int |p(\bar{z}|s',a') - q(\bar{z}|s',a')|^n \, d\bar{z} \right]$$

$$\leq \gamma \sup_{s,a} \sup_{s',a'} \left( \int |p(\bar{z}|s',a') - q(\bar{z}|s',a')|^n \, d\bar{z} \right)$$

$$= \gamma \sup_{s',a'} \int |p(\bar{z}|s',a') - q(\bar{z}|s',a')|^n \, d\bar{z}$$

$$= \gamma \sup_{s,a} L_n(p(\cdot|s,a), q(\cdot|s,a))^n$$

$\square$

**Proof Theorem 4.5.**  Let us apply the operator $\mathcal{P}^\pi$ to the distribution $q^\pi$

$$\mathcal{P}^\pi q^\pi(z|s,a) = \mathcal{P}^\pi \int h(z|\bar{s},\bar{a}) d^{\pi,\gamma}(\bar{s},\bar{a}|s,a) \, d\bar{s} \, d\bar{a}$$

$$= \mathop{\mathbb{E}}_{\substack{s' \sim p(\cdot|s,a) \\ a' \sim \pi(\cdot|s')}} \left[ (1-\gamma)h(z|s',a') + \gamma \left( \int h(z|\bar{s},\bar{a}) d^{\pi,\gamma}(\bar{s},\bar{a}|s',a') \, d\bar{s} \, d\bar{a} \right) \right]$$

$$= \int h(z|\bar{s},\bar{a}) \left( \mathop{\mathbb{E}}_{\substack{s' \sim p(\cdot|s,a) \\ a' \sim \pi(\cdot|s')}} [(1-\gamma)\delta_{\bar{s},\bar{a}}(s',a') + \gamma d^{\pi,\gamma}(\bar{s},\bar{a}|s',a')] \right) d\bar{s} \, d\bar{a}$$

$$= \int h(z|\bar{s},\bar{a}) \left( (1-\gamma)\pi(\bar{a}|\bar{s})p(\bar{s}|s,a) + \gamma \mathop{\mathbb{E}}_{\substack{s' \sim p(\cdot|s,a) \\ a' \sim \pi(\cdot|s')}} [d^{\pi,\gamma}(\bar{s},\bar{a}|s',a')] \right) d\bar{s} \, d\bar{a}$$

$$= \int h(z|\bar{s},\bar{a}) d^{\pi,\gamma}(\bar{s},\bar{a}|s',a') \, d\bar{s} \, d\bar{a}$$

$$= q^\pi(z|s,a) \,.$$

$\square$

## C  SOFT ACTOR-CRITIC WITH CONDITIONAL VISITATION MEASURE

In the following, we adapt soft actor-critic (Haarnoja et al., 2018b), itself an adaptation of off-policy actor-critic (Degris et al., 2012), according to the procedure from Section 4. In essence, soft actor-critic estimates the state-action value function with a parameterized critic $Q_\phi$, which is learned using expected SARSA (sometimes called generalized SARSA), and updates the parameterized policy $\pi_\theta$ with approximate policy iteration (i.e., off-policy policy gradient), all based on one-step transitions stored in a replay buffer $\mathcal{D}$. The actor and critic loss functions are furthermore extended with the log-likelihood of actions weighted by the parameter $\lambda_{SAC}$, therefore called soft and considered a MaxEntRL algorithm using the entropy of policies as intrinsic reward. In the particular case where $\lambda$ equals zero, the algorithm boils down to a slightly revisited implementation of off-policy actor-critic.

Soft actor-critic is adapted to MaxEntRL with the intrinsic reward function defined in Section 3.1, as follows. First, $N$-step transitions are stored in the buffer $\mathcal{D}$ instead of one-step transitions. Second, the conditional feature distribution is estimated with a function approximator $q_\psi$ and learned with stochastic gradient descent. Third, at each iteration of the critic updates, the reward provided by the MDP is extended with the intrinsic reward.

Formally, the parameterized critic $Q_\phi$ is iteratively updated performing stochastic gradient descent steps on the loss function

$$\mathcal{L}(\phi) = \mathbb{E}_{s_t, a_t \sim \mathcal{D}}\left[\left(Q_\phi(s_t, a_t) - y\right)^2\right] \tag{15}$$

$$y = R(s_t, a_t) + \lambda R^{int}(s_t, a_t) + \gamma\left(Q_{\phi'}(s_{t+1}, a_{t+1'}) - \lambda_{SAC}\log\pi_\theta(a_{t+1'}|s_{t+1})\right), \tag{16}$$

where $a_{t+1'} \sim \pi_\theta(\cdot|s_{t+1})$, and where $\phi'$ is the target network parameter.

Furthermore, the policy $\pi_\theta$ is updated performing gradient descent steps on the loss function

$$\mathcal{L}(\theta) = -\mathbb{E}_{s_t, a_t \sim \mathcal{D}}\left[\log\pi_\theta(a_{t'}|s_t)A(s_t, a_{t'})\right] \tag{17}$$

$$A(s_t, a_{t'}) = Q_\phi(s_t, a_{t'}) - \lambda_{SAC}\log\pi_\theta(a_{t'}|s_t), \tag{18}$$

where $a_{t'} \sim \pi_\theta(\cdot|s_t)$.

Algorithm 1 summarizes the learning steps during each iteration[1]. It differs slightly from the original soft actor-critic (Haarnoja et al., 2018b). The loss equation equation 17 is based on the log-trick instead of the reparametrization trick, the expected SARSA update in equation equation 15 is approximated by sampling, and a single value function is learned, as implemented in CleanRL (Huang et al., 2022). These changes are of minor importance in our experiments.

---

[1]Implementation details at `https://github.com/anonymized-for-review`

**Algorithm 1** SAC with conditional visitation measure for exploration

---

Initialize the policy $\pi_\theta$, the soft critic $Q_\phi$, and the conditional feature model $q_\psi$
Initialize the critic target $Q_{\phi'}$ and visitation target $q_{\psi'}$
Initialize the replay buffer with random $N$-step transitions
**while** Learning **do**
   Sample transitions from the policy $\pi_\theta$ and add them to the buffer
   **while** Update the visitation model **do**
      Sample a batch of $N$-step transitions from the buffer
      Update the visitation model
   **end while**
   **while** Update the critic **do**
      Sample a batch of $N$-step transitions from the buffer (use only the 1-step transitions)
      For each element of the batch sample $z_t \sim q^\pi(\cdot|s_t, a_t)$
      Estimate the intrinsic reward $R^{int}(s_t, a_t) = \log q^*(z_t) - \log q^\pi(z_t|s_t, a_t)$
      Perform a stochastic gradient descent step on $\mathcal{L}(\phi)$
   **end while**
   Sample a batch of $N$-step transitions from the buffer (use only the 1-step transitions)
   Perform a stochastic gradient descent step on $\mathcal{L}(\theta)$
   Update the target parameters with Polyak averaging
**end while**

---

# D EXPERIMENTAL RESULTS

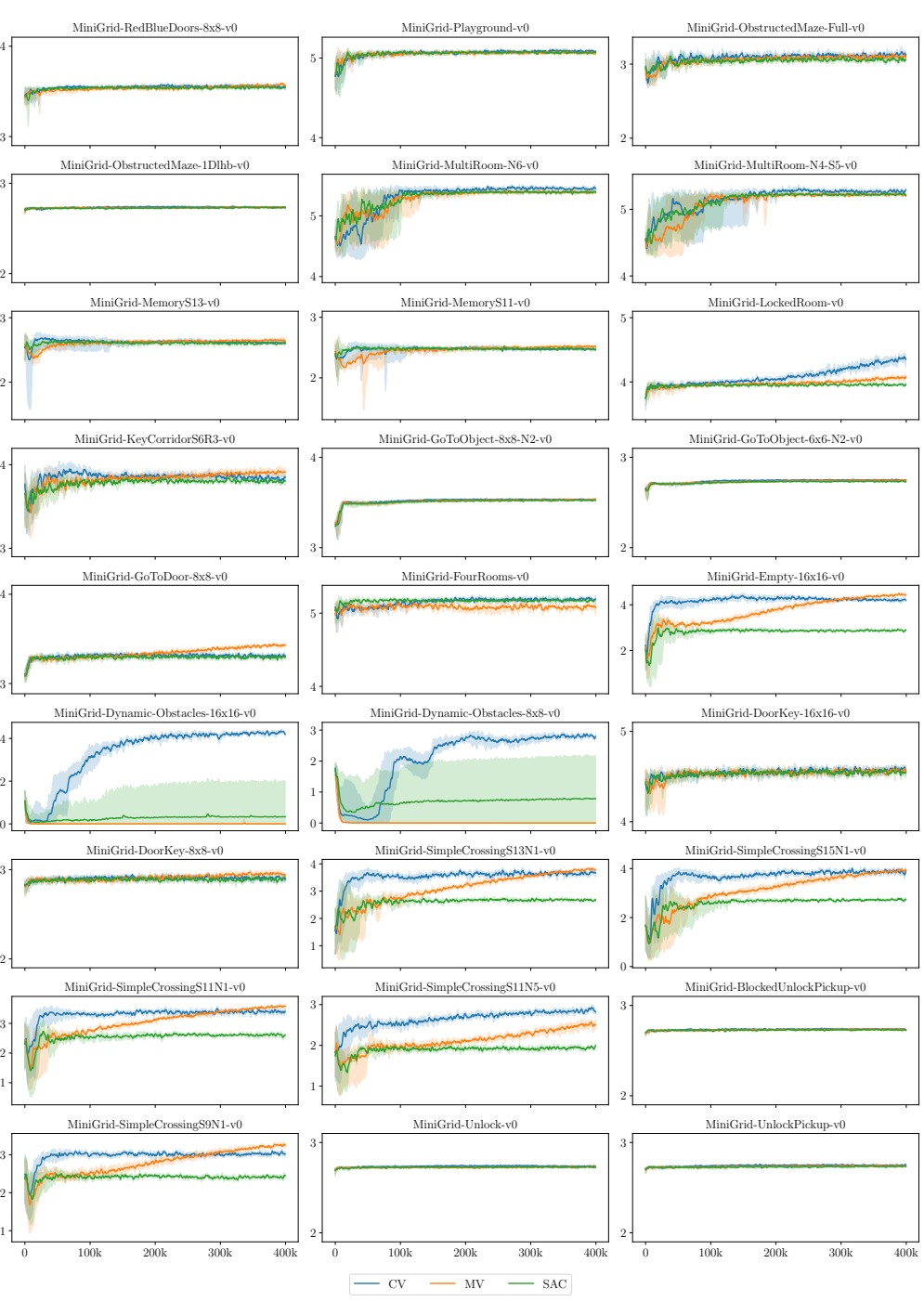

Figure 1: Evolution of the entropy of the discounted visitation probability measure of the position of the agent on the grid when computing exploration policies (i.e., when neglecting the rewards of the MDP). The entropy is computed empirically with Monte Carlo simulations. For each iteration, the interquartile mean over 15 runs is reported, along with its $95\%$ confidence interval.

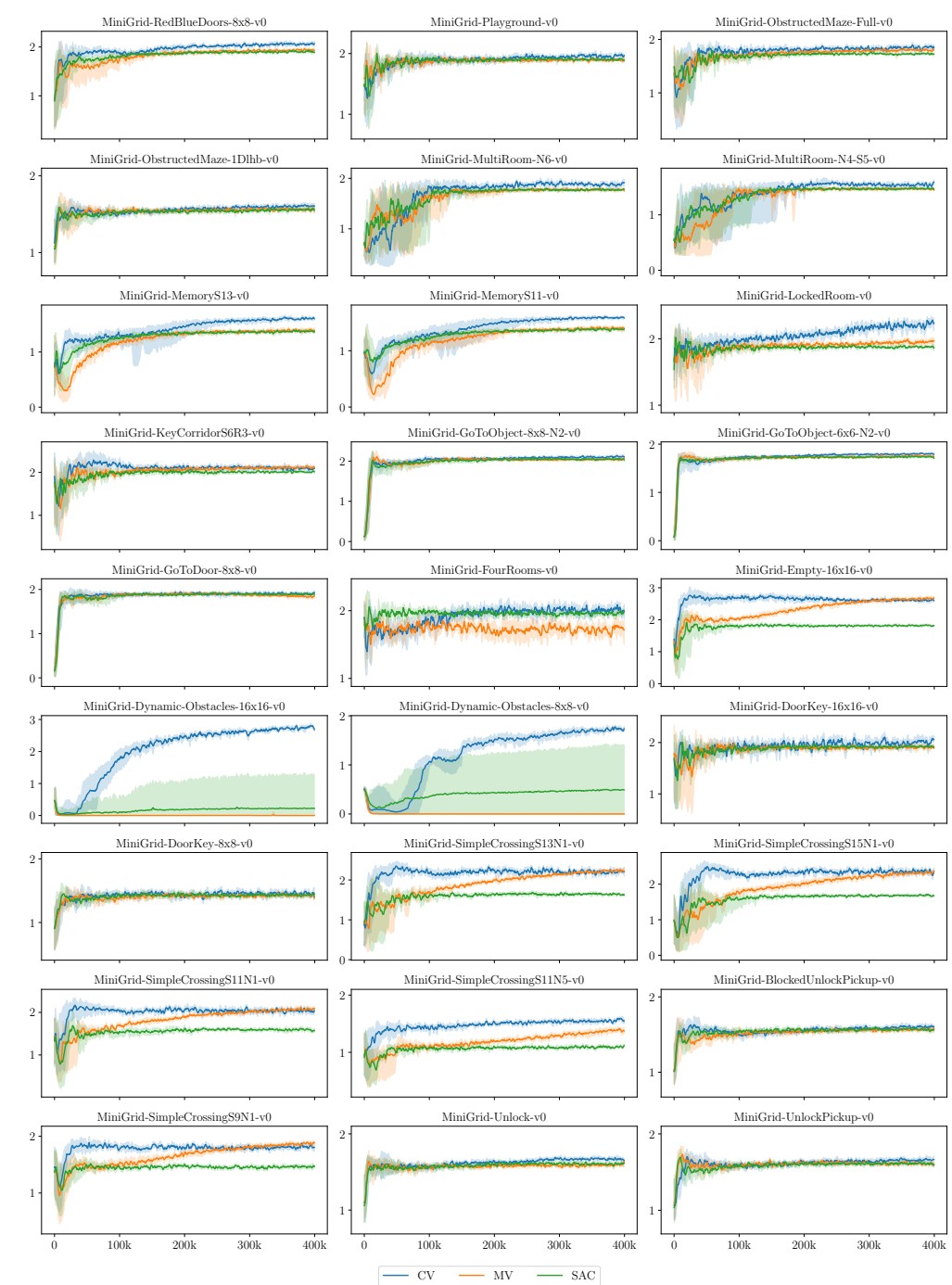

Figure 2: Evolution of the conditional entropy of the discounted visitation probability measure of the position of the agent on the grid when computing exploration policies (i.e., when neglecting the rewards of the MDP). The entropy is computed empirically with Monte Carlo simulations. For each iteration, the interquartile mean over 15 runs is reported, along with its $95\%$ confidence interval.

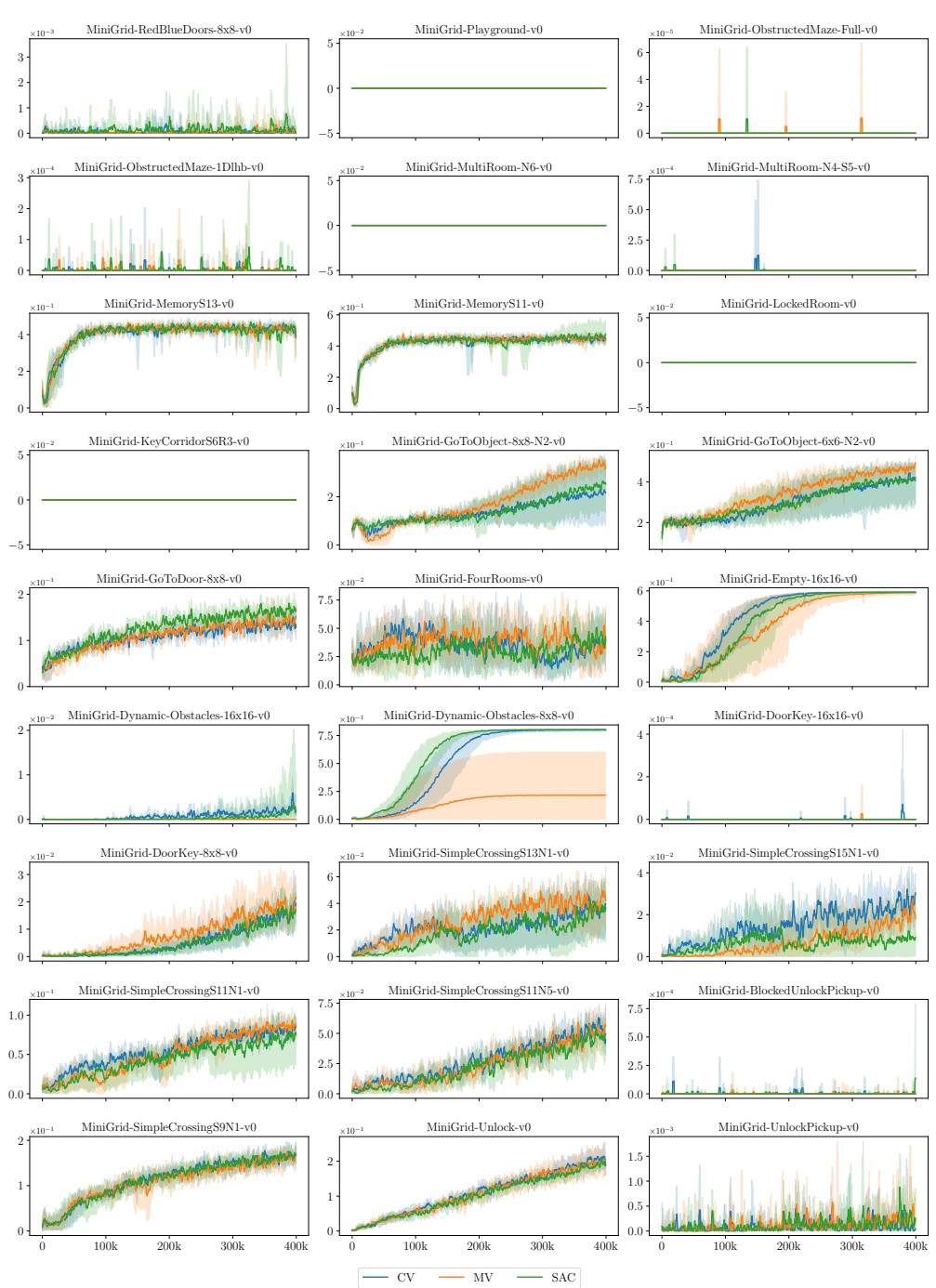

Figure 3: Expected return during the policy optimization. The expectation is computed empirically with Monte Carlo simulations. For each iteration, the interquartile mean over 15 runs is reported, along with its 95% confidence interval.

# E HYPERPARAMETERS EXPERIMENTS

In this section, we detail implementation details for reproducing the experiments.

In practice, the agent observes an image that is processed by a convolutional neural network into a state feature. The policy $\pi_\theta$ is a forward network that processes this feature and that outputs a categorical distribution over the action representation. The critic $Q_\phi$ is a neural network that takes as input the concatenation of the state feature and a linear projection of the action representations and outputs a scalar. In CV, the distribution model $q_\psi$ is also a neural network that takes the same input as the critic $Q_\phi$ and outputs a categorical distribution over a one-hot-encoding representation of positions. In MV, the visitation distribution model $q_\psi$ is a marginal distribution over the same one-hot-encoding representation. All previous models are learned independently using Adam optimization (Kingma & Ba, 2014).

Table 1 summarizes the hyperparameters used for the experiments.

Table 1: Hyperparameters

| Parameter | Value |
|---|---|
| Convolutional layer | 2 |
| Neurons for each convolutional layer | 32 |
| State-feature size | 256 |
| Forward layers policy | 2 |
| Forward layers critic | 2 |
| Neurons forward layers | 256 |
| Action-projection size critic | 256 |
| Learning rate policy | $10^{-6}$ |
| Learning rate critic | $10^{-5}$ |
| Maximum trajectory length | 200 |
| Buffer size | 100000 |
| Batch size | 128 |
| Critic target update weight $\tau$ | 0.01 |
| Discount factor $\gamma$ | 0.98 |
| SAC $\lambda_{SAC}$ | 0.001 |
| Forward layers visitation model | 2 |
| Neurons forward layers | 256 |
| Action-projection size visitation model | 256 |
| Learning rate visitation model | $10^{-6}$ |
| MaxEntRL $\lambda$ | 0.001 |
| Density model target update weight $\tau$ | 0.01 |

