# OpenReview forum: "Maximum-Entropy Exploration with Future State-Action Visitation Measures"
_ICLR.cc/2026/Conference — Submitted to ICLR 2026_

### Official Review · Reviewer_jJby · 2025-10-27

**Soundness:** 3
**Presentation:** 3
**Contribution:** 3
**Rating:** 6
**Confidence:** 4

**Summary:**

This paper proposes a new maximum entropy reinforcement learning (MaxEntRL) objective that encourages exploration by maximizing the entropy of conditional discounted future visitation distributions over state-action features. Unlike classical MaxEntRL approaches that either maximize policy entropy or the entropy of marginal visitation measures, this work focuses on the conditional visitation measure starting from each state-action pair. The new objective is shown to be a lower bound on the standard marginal visitation entropy objective, thereby retaining desirable exploration incentives while potentially simplifying computation. The conditional visitation distribution is the unique fixed point of a contraction operator, enabling off-policy estimation of intrinsic rewards via N-step bootstrapping rather than expensive on-policy rollouts. On the algorithmic side, the authors adapt soft actor-critic  to include this new intrinsic reward, learned through an auxiliary visitation model trained with a cross-entropy objective. Empirical evaluation on MiniGrid environments compares three exploration strategies-policy entropy, marginal visitation entropy, and the proposed conditional visitation entropy — using both exploration metrics (feature entropy over episodes and within trajectories) and task performance. Results show that the new objective improves within-trajectory exploration and sometimes matches or outperforms marginal visitation methods in terms of entropy, while maintaining similar control performance.

**Strengths:**

1. The paper is well-structured, with clear definitions of visitation distributions and intrinsic reward functions.
2. The lower bound theorem (Thm 3.2) provides a theoretical justification,  relating the conditional and marginal objectives.
3.  The contraction property (Thms 4.2–4.5) is a strong technical result that underpins the off-policy learning advantage.
4. The approach provides a middle ground between purely on-policy entropy methods and more complex state-visitation-based objectives.
5. Off-policy compatibility addresses a well-known bottleneck of previous state-visitation entropy methods, which are typically on-policy and sample-inefficient.

**Weaknesses:**

1. All experiments are conducted on MiniGrid, which is discrete, small-scale, and relatively simple compared to widely used continuous control benchmarks (e.g., Mujoco, DM Control Suite). It is unclear how the method scales in high-dimensional continuous state-action spaces, especially regarding density estimation.
2.  The implementation drops importance weighting and other terms, introducing bias in the visitation model estimation. The practical impact of this bias is not fully analyzed.
3. While exploration improves (particularly within trajectories), final task returns are comparable to baseline MaxEntRL methods. t remains to be shown whether improved exploration translates into clear performance benefits in more challenging settings.
4.  While the method is simpler than marginal visitation estimation, the additional training of a visitation model still adds overhead. No detailed runtime or sample efficiency comparison is provided.

**Questions:**

1. How does the method perform on environments with large or continuous state-action spaces (e.g., HalfCheetah, Ant)? Which parts of the algorithm (e.g., density estimation, model learning) would need to be adapted ?
2. The paper shows the theoretical lower bound but does not empirically analyze how close the conditional and marginal objectives are during training. Can you provide such an analysis ?
3. How significant is the approximation bias introduced by neglecting importance weights in Eq. (12) ? Could variance reduction techniques help retain theoretical guarantees ?
4. Could the feature space Z be learned jointly with the policy (e.g., using contrastive learning or predictive representations) instead of being predefined?
5. How does the training time and sample efficiency compare to marginal visitation methods, especially when scaling up N or the environment complexity?

---

> ### Author Response · Authors · 2025-11-27
>
> Thank you for reading our paper and for your detailed review. We address your concerns below and explain why we believe they are resolved. In light of these clarifications, we kindly ask you to reconsider your evaluation and to increase your score in support of acceptance.
>
> > All experiments are conducted on MiniGrid. It is unclear how the method scales in high dimensional continuous spaces and for density estimation.
>
> MiniGrid environments are challenging from an exploration viewpoint because of their very large discrete state spaces. This makes them well suited to test exploration strategies while keeping visitation distributions interpretable. As discussed in the paper, the objective and algorithm extend to continuous spaces by slightly changing the density model. For instance, one can use normalizing flows with tractable log likelihoods or optimize an ELBO when the likelihood is not available in closed form. We chose to focus on discrete environments to validate the main hypotheses and leave a continuous control study as future work. We will clarify this adaptation path in the revised version.
>
> > The implementation drops importance weighting and other terms, introducing bias in the visitation model estimation. The practical impact of this bias is not fully analyzed.
>
> Removing importance weights introduces a dependence on the distribution of states and actions stored in the replay buffer. This approximation is common in practice when replay is used with methods that ideally require on policy data, and is also related to standard TD learning with replay. In our case, the bootstrapping structure of the conditional objective reduces the need for accurate full trajectory corrections compared to marginal visitation entropy. Empirically we observed that this approximation leads to more stable learning. We will add a short discussion in the paper to make this trade off explicit.
>
> > While exploration improves, final task returns are comparable to baseline MaxEnt RL methods. It remains to be shown whether improved exploration gives clear performance benefits in more challenging settings.
>
> Many works on exploration study reward free settings where the quality of exploration is the primary object of interest. Our experiments show that the conditional objective yields higher conditional entropy and thus richer feature coverage within individual trajectories while keeping marginal entropy and returns comparable to marginal visitation methods. It also converges faster in the reward free setting. We see this as evidence that the objective targets a different, and in some settings more relevant, notion of exploration. More broadly, this touches the open question of what constitutes a good exploration objective. Our contribution is to highlight a principled alternative with distinct properties rather than to claim universal performance gains.
>
> > The method adds overhead due to training a visitation model. No detailed runtime or sample efficiency comparison is provided.
>
> Training the visitation model indeed adds computational overhead, mostly for the largest environments. In terms of complexity, this is comparable to learning a transition model in model-based RL, since both involve an additional network updated from replay. Our experiments focus on sample efficiency, as is standard too when comparing model-based and model-free methods. We will clarify this trade-off and add more explicit comments on runtime in the experimental section.
>
> [Follow-up below.]

---

> ### Author Response · Authors · 2025-11-27
>
> [Response part 2]
>
> > How does the method perform on environments with large or continuous state action spaces. Which parts of the algorithm would need adaptation.
>
> Some of the MiniGrid tasks already have very large state spaces. We have not yet run experiments on continuous control benchmarks such as HalfCheetah or Ant. As noted above, the main adaptation lies in the density estimator, for example, using a normalizing flow or a model trained with an ELBO. In such domains, representation learning becomes central, and the exploration problem interacts strongly with the choice of features. We see this as a promising but nontrivial extension and will clarify which components need to change.
>
> > The paper shows the theoretical lower bound but does not empirically analyze how close the conditional and marginal objectives are during training.
>
> Estimating the full expected conditional entropy is challenging. In the paper we report both marginal entropies and conditional entropies given the initial state, which partially reflect the gap between the two objectives. We also observe that the marginal entropy of our method is slightly lower during training but seems to match the marginal method at convergence, which is consistent with the lower bound interpretation. We agree that a more direct empirical analysis of this gap would be interesting and consider it an avenue for follow up work. We will clarify these observations in the text.
>
> > How significant is the approximation bias from neglecting importance weights in Eq. (12). Could variance reduction help retain guarantees.
>
> A precise theoretical characterization of this bias is outside the scope of the paper and appears technically difficult. Empirically, we find that the bias is mitigated by the continual refresh of the replay buffer, which keeps the behavior and target policies reasonably close. Similar approximations are widely used when learning visitation based quantities from replay, and also in TD methods. We agree that combining our conditional objective with more advanced variance reduction or off policy correction techniques is an interesting research direction and will mention this as future work.
>
> > Could the feature space Z be learned jointly with the policy rather than predefined.
>
> Yes, this is an important open question: what to explore (state, actions, features?). One can typically either explore a learned representation of the state, for example, from an encoder trained with contrastive or predictive objectives, or focus exploration on features that are predictive of reward, which is often motivated by the noisy TV argument. Both approaches raise nontrivial representation learning issues. In this paper we deliberately chose a handcrafted feature space, the agent position, which is reward predictive by construction in MiniGrid. This removes confounding effects from representation learning and isolates the effect of the exploration objective. We will clarify this design choice and discuss the possibility of jointly learning Z as future work.
>
> > How does training time and sample efficiency compare to marginal visitation methods when scaling up N or environment complexity.
>
> Our experiments indicate that the conditional objective is generally more sample efficient, which we attribute to the off policy learning enabled by its bootstrapping structure. Varying N induces a trade off. Larger N is beneficial with high discount factors since it avoids excessive bootstrapping, but it can also introduce instability through the use or neglect of importance weights. In practice we selected an intermediate value of N that provided a good compromise. We will expand the discussion of this trade off and its empirical impact in the experimental section.
>
> We appreciate your constructive feedback and believe that with the clarifications above the contribution and limitations of the paper are accurately represented. We therefore respectfully ask you to reconsider your score and support acceptance.

---

### Official Review · Reviewer_qoEK · 2025-10-28

**Soundness:** 2
**Presentation:** 1
**Contribution:** 1
**Rating:** 2
**Confidence:** 4

**Summary:**

1. The author replace the entropy term H during full trajectories into H during future time steps.
2. Use this entropy term as intrinsic reward and propose a new method
3. "All methods lead to similar control performance on the considered benchmarks."

**Strengths:**

The author provide a theoretical analysis for the proposed method.

**Weaknesses:**

The paper is poorly written and contains unclear language and disorganized structure.

1. **Use of Eq (4) for Intrinsic Reward Formulation:**
   Why is Eq (4) chosen as the intrinsic reward formulation? How can this formulation be equivalent to the MaxEntRL or the particle-based entropy estimation method (Hazan et al., 2019) mentioned in line 14?

2. **Explanation of Variables and Distributions:**
   The explanation of variables and distributions is inadequate. What do \( \bar{s} \) and \( \bar{a} \) represent? What is the feature function \( h \)?
   Furthermore, the term \( q_\pi(z|s, a) \) appears multiple times in Section 2.2, but its definition is only provided in Section 3. It seems these are different entities, which causes confusion. It would be helpful to provide the definition earlier and give an intuitive explanation for clarity.

3. **Experimental Results on Grid Experiment:**
   The experimental results on the Grid experiment do not demonstrate superior performance. The results do not provide sufficient evidence to support the claims of the paper.

4. **Baseline Comparison:**
   The baseline includes only the SAC algorithm, but there are many exploration methods that should be considered, such as ICM and NGU. For entropy-driven methods, the particle-based entropy estimation method should also be included as a baseline.

Overall, the paper is poorly written, making it difficult to extract useful information. The method, algorithm, and contributions lack clarity. I recommend rejecting this paper and suggest that it be revised and resubmitted later after addressing the above issues.

**Questions:**

See Above.

---

> ### Author Response · Authors · 2025-11-27
>
> Thank you for accepting to review our paper. Unfortunately, your summary does not match the actual content and several remarks are factually incorrect or already addressed in the main text. We kindly ask you to reread the paper more carefully in order to assess the contribution fairly. We are happy to improve clarity where needed, but we need concrete pointers to specific passages.
>
> > Use of Eq. (4) for intrinsic reward formulation
>
> Eq. (4) defines a generic Maximum Entropy RL objective. The paragraph line 154 following Eq. (4) explains how different choices of the distribution $q_\pi$ recover existing MaxEnt objectives and how our conditional visitation entropy objective is obtained as a specific instance. The question you raise is explicitly answered in that paragraph.
>
> > Explanation of variables and distributions
>
> The core design choice in MaxEnt RL is what space to explore. Common choices are the state space or the state action space. Our formulation generalizes this by introducing a feature map $h$ that specifies which features of state action pairs are explored. This is defined explictely line 139 as a distribution over features induced by $h$. We then give concrete examples, including the standard marginal visitation entropy objective and our new conditional visitation entropy objective.
>
> > Experimental results on grid experiment
>
> The experiments show that our method achieves higher conditional entropy and richer feature coverage within individual trajectories, while maintaining strong marginal entropy and similar return. They also show faster convergence in the reward free setting, which we attribute to the off policy learning enabled by our conditional objective and fixed point structure. These points are described in the experimental section and supported by the plots.
>
> > Baseline comparison
>
> The exploration baselines are marginal state (or feature) entropy and action entropy. SAC serves as the common RL backbone, not as an exploration baseline. This choice isolates the effect of the exploration objective, which is the focus of the paper. We will clarify this in the introduction and experimental section to avoid confusion.
>
> Given that most of your concerns stem from misunderstandings that can be resolved by a more careful reading and minor clarifications in the text, we respectfully ask you to reconsider your evaluation and to increase your score in support of acceptance.

---

### Official Review · Reviewer_kSFD · 2025-10-30

**Soundness:** 3
**Presentation:** 3
**Contribution:** 2
**Rating:** 2
**Confidence:** 4

**Summary:**

The paper proposes a new intrinsic reward for exploration in RL, which combines the commonly used action entropy and another entropy term on the conditional states visitation. Implementing this intrinsic reward requires to model the state visitation, here obtained through a TD-like methodology in a discounted setting. The intrinsic reward can be incorporated into any discounted RL algorithm, e.g., SAC, for improved exploration on the state space. The resulting method is evaluated against vanilla SAC and another baseline maximizing the entropy of the marginal state visitation in some sparse-rewards environment from MiniGrid.

**Strengths:**

- The paper provides an elegant unification of different "MaxEntRL" approaches by formalizing the intrinsic reward with a separate feature space, which can alternatively be the action space for standard action entropy incentives or the state space for state entropy exploration;
- The paper provides an original version of the state entropy exploration objective that only looks at the entropy of the future discounted state (or state-action) distribution conditioned on the current state;
- The paper provides a few theoretical results showing that the introduced intrinsic reward is the fixed point of a contractive operator and that it constitutes a lower bound of the more common marginal state visitation entropy objective.

**Weaknesses:**

- The paper seems to mischaracterize existing state entropy algorithms as inherently on-policy. While several of the existing implementations are used on-policy, they can be easily adapted to work off-policy;
- After reading the paper and looking at experimental results, why conditional entropy shall be preferred to marginal entropy is largely unclear to me;
- The paper does not much to clarify why MiniGrid has been chosen to compare the performance of conditional visitation entropy with prior work, which typically consider much more challenging domains, such as Mujoco or Atari;
- The paper circumvent the page limit by placing all the plots in the appendix, although almost a full page of the main paper is left blank, so that I would consider this mostly as a poor formatting decision rather than a style violation.

**EVALAUTION**

While the paper is original and promising, I think the paper lacks strong conceptual and empirical ground to support the proposed intrinsic reward. Advancements over prior works shall be clarified to meet the bar for acceptance.

**Questions:**

Other than the weaknesses mentioned above, I have two additional comments the authors may consider in their response.

1) Is conditional visitation better than marginal in some sense?

After reading the paper, I am not convinced that conditional visitation is better than marginal for the purpose of pre-training a policy to be used for efficient learning of a downstream task. If the abstract objective is to induce even visitation of the states (and actions), I would argue that the past also matters. Let us look at this simple example: We have two rooms (left and right) connected by a corridor. The agent starts in the middle of the corridor with the goal to maximize the conditional entropy. Let's say the agent start exploring the left room before going back to the corridor. Now, since the objective is conditioned on the current state and action and only looks at the future, visiting the right room or the left room again is equivalent. Thus, a policy that visits the left room repeatedly is optimal for the conditional objective. Instead, a policy maximizing the marginal state visitation, will necessarily randomly choose between going left or right when in the middle. If the policy is history-based, it shall choose to go left and right depending on what has been previously explored. In this example, the marginal entropy looks much more aligned with the abstract exploration objective. I am wondering if there are other examples where the conditional entropy is better.

2) On-policy vs off-policy methods

The marginal state entropy can also be optimized off-policy. In various implementations, e.g., see Abbeel et al., 2021 or Seo et al., 2021, the objective is cast into an intrinsic reward that can be optimized with SAC, more or less as it is done in the experiments here. So i do not understand why the paper claims that the conditional visitation opens the door to off-policy methods while marginal state entropy doesn't. Perhaps a more supported claim would be to note that conditional entropy is the fixed point of a contractive operator, while a similar result is not known for marginal state entropy.

---

> ### Author Response · Authors · 2025-11-27
>
> Thank you for reading and reviewing our paper. The main issues you raise concern the relation between on policy and off policy learning for state entropy methods and the specific benefit of our conditional visitation entropy objective. We clarify these points below and explain why they support acceptance.
>
> > The paper seems to mischaracterize existing state entropy algorithms as inherently on policy. While several existing implementations are used on policy, they can be adapted to work off policy.
>
> It is useful to distinguish two aspects. First, the control backbone, for instance SAC, which can be off policy. Second, the way the visitation distribution used for the intrinsic reward is learned. In tabular settings, marginal visitation can be estimated off policy by learning a full transition model, as in the original state entropy works. In the function approximation setting, existing methods update the entropy estimator from (near) on policy rollouts, and we are not aware of practical off policy counterparts. Our contribution is a conditional objective whose entropy estimator satisfies a contraction property and can be updated with an N step bootstrapping scheme from replay. In principle one could use full trajectory importance sampling for marginal entropy, but this would require long horizon corrections and is known to have prohibitive variance, which is why it is not used in practice. We will clarify this distinction in the paper.
>
> > After reading the paper and looking at experimental results, why conditional entropy should be preferred to marginal entropy is unclear.
>
> Our aim is not to claim that conditional entropy is always better than marginal entropy, but that it optimizes a different exploration objective with attractive properties. Empirically, our method achieves similar marginal entropy and return, yet improves conditional entropy and thus feature coverage within trajectories. In several MiniGrid tasks it also converges faster in the reward free setting. Finally, in continuing problems, future visitation given the current state action is likely more relevant than global marginal visitation. Our conditional objective targets these aspects while preserving good marginal coverage, which we believe is a meaningful alternative.
>
> > The paper does not clarify why MiniGrid has been chosen, while prior work often uses domains such as Mujoco or Atari.
>
> MiniGrid environments are known to be exploration hard because of their very large combinatorial state spaces. This makes them well suited to stress test exploration objectives. They also allow direct inspection of visitation distributions and conditional coverage, which is difficult in high dimensional visual domains. As discussed in the paper, our objective and algorithm extend to continuous spaces, and applying them to Mujoco or Atari is a natural direction for future work, but not required to validate the conceptual contribution.
>
> > The paper circumvents the page limit by placing plots in the appendix, although almost a full page of the main paper is left blank.
>
> We agree that this formatting choice is not ideal. In the revised version we will move representative learning curves and exploration metrics to the main text so that the reader can more easily follow the experimental section. This is a straightforward presentation fix and does not affect the technical content.
>
> > Question 1. Is conditional visitation better than marginal in some sense?
>
> This relates to the broader open question of what makes a good exploration objective. Our contribution is to propose and analyze a new objective rather than to prove that it dominates marginal entropy in all settings. We provide several arguments in its favor. First, we show empirically that it yields higher conditional entropy and thus richer feature coverage within individual trajectories while preserving strong marginal coverage. Second, in continuing tasks we never revisit past states, so the distribution of future states given the current state action is the relevant quantity for exploration. Our objective is defined as an expectation over such conditionals, so it does not suffer from the corridor failure mode suggested in your example. In fact, corridor type environments are present in the MiniGrid suite we consider, and our method performs well there.
>
> > Question 2. On policy vs off policy methods.
>
> The key difficulty for state entropy methods is how to learn the intrinsic reward itself. For marginal entropy with function approximation, existing approaches rely on (near) on policy data. By contrast, our conditional objective leads to a Bellman type fixed point for the entropy related quantity. This enables an off policy N step update from replay, so that both the intrinsic reward and the policy are learned off policy in a principled way.
>
> In light of these clarifications, we kindly ask the reviewer to increase their score and support acceptance of the paper.

---

> > ### Comment · Reviewer_kSFD · 2025-11-27
> >
> > Dear authors,
> >
> > Thank you for your clarifications.
> >
> > As a brief follow-up:
> > - (Off-policy/on-policy) One can estimate the intrinsic reward for the marginal entropy as $\log d (s, kNN(s))$ and then learn from off-policy data. I doubt it is a good idea, but why this does not account as off-policy method is note clear to me. Perhaps what the authors meant to say is that their approach can go "more" off-policy (works better when distribution shift is large). Empirically testing this claim could help.
> > - (Marginal vs conditional) The motivation for which "we never revisit past states" sounds weak to me. Even though the physical state may be unique, we still want to consider states to be "similar" at higher level of abstraction.
> >
> > Best regards,
> >
> > Reviewer kSFD

---

> ### Author Response · Authors · 2025-11-27
>
> Thank you for your fast response and for the additional clarifications.
>
> Regarding the off policy aspect, we agree that in principle one can define an intrinsic reward based on marginal entropy and then optimize it with an off policy control algorithm. Our point is about how the visitation model itself is trained. From our understanding of the literature, with a KNN estimator, one essentially fits the density of the states and actions present in the replay buffer, which reflects the behavior distribution. Using this estimator for policies that are far from the data-generating policy would require some form of correction. In our approach, the conditional quantity satisfies a Bellman-type fixed point, so we can update it with N step bootstrapping from replay in a way that is more tolerant to distribution shift.
>
> Regarding the marginal versus conditional motivation, we emphasize that the phrase “we never revisit past states” is an abuse of language. The key point is that our objective encourages, in each state, a diverse distribution over future features, rather than rewarding past novelty too. It implies that the future states are not preferred when less likely visited in the past, or inversely, as with the marginal effect. Past states still matter with the expectation over these intrinsic rewards. In many continuing tasks we see this future-oriented criterion as natural, but we do not claim it is universally superior. As noted in the rebuttal, identifying the best exploration objective is an open question, and we present conditional visitation entropy as a principled alternative with different trade-offs.
>
> Best regards

---

### Official Review · Reviewer_2AbX · 2025-10-31

**Soundness:** 2
**Presentation:** 2
**Contribution:** 2
**Rating:** 2
**Confidence:** 2

**Summary:**

The paper introduces a new MaxEntRL intrinsic reward based on the conditional state-action visitation probability and the conditional state visitation probability.  This objective extends classical one where Shannon entropy of policy is considered (feature space is action space). The paper proposes an off-policy learning algorithm integrating this reward into Soft Actor-Critic (SAC). Empirical results on MiniGrid environments, comparing three exploration strategies (policy entropy, marginal visitation, conditional visitation) are provided.

**Strengths:**

Integrating visitation distributions into MaxEntRL is an interesting idea. There are theoretical results (e.g contractive properties and KL lower bounds) and numerical experiments on MiniGrid environment.

**Weaknesses:**

1. The function $h$ is central but under-specified. How to choose or parameterize it for high-dimensional states is unclear
2. Only small, discrete MiniGrid environments are tested. No continuous-control task are considered.  The current experiments rely on hand-crafted discrete features (agent position) limiting generality.
3. There is no intuitive explanation of why the lower bound of Th. 3.2  is meaningful for exploration or what properties it preserves. What is L in this theorem. It would be good to specify if it is an absolute constant.
4. The paper doesn't provide any theoretical performance or sample-complexity advantage from using different maximum entropy objectives. All the theoretical results obtained in the paper are generally trivial.

**Questions:**

1. There are no assumptions in the formulations of the main theorems. Do I understand correctly that the theorems are always fulfilled for any spaces of states and actions?
2. The authors report that “all methods lead to similar control performance.” What is the practical impact in this case?

**Details Of Ethics Concerns:**

-

---

> ### Author Response · Authors · 2025-11-27
>
> Thank you for reviewing our paper. We believe some key aspects of our contribution were overlooked, and we hope the answers below clarify them. The goal of the paper is not primarily to extend policy entropy regularization, but to propose an alternative to state-action (feature) entropy exploration, which is widely used despite limited theoretical guarantees.
>
> Our method introduces an exploration objective based on the entropy of the conditional future state action visitation distribution. The theoretical results show that this objective preserves good marginal coverage while improving conditional coverage within individual trajectories. Depending on the type of exploration desired, our method can therefore be preferable while achieving similar control performance.
>
> > The function h is central but under specified. How to choose or parameterize it for high dimensional states is unclear.
>
> The choice of h is a general challenge for all entropy based exploration methods and is not specific to our approach. In practice, h can be fixed a priori (for instance using the raw state in low dimensional domains), based on expert knowledge (as in our MiniGrid experiments, where we use agent position), or obtained from a learned encoder. We explicitly focus on the expert knowledge setting and do not claim to solve the general representation learning problem, which is orthogonal to the proposed conditional entropy objective.
>
> > Only small, discrete MiniGrid environments are tested. No continuous control tasks are considered. The current experiments rely on hand crafted discrete features (agent position), limiting generality.
>
> MiniGrid environments are known to be particularly challenging from an exploration perspective due to their very large state spaces, which makes them well suited to evaluate exploration strategies. We chose them because they allow us to interpret and visualize visitation distributions and conditional coverage. As discussed in the paper, the proposed objective and algorithm extend to continuous spaces. Studying continuous control domains is an interesting direction for future work but is not required to validate the conceptual contribution.
>
> > There is no intuitive explanation of why the lower bound of Th. 3.2 is meaningful for exploration or what properties it preserves. What is L in this theorem. It would be good to specify if it is an absolute constant.
>
> Theorem 3.2 connects our conditional entropy objective to the standard marginal visitation entropy objective. Under the stated assumptions, it shows that at optimality the conditional objective forms a lower bound on the marginal objective up to an additive constant L that depends on the feature map h and the environment but is independent of the policy. This explains why the marginal entropy remains high in our experiments. We will clarify in the camera ready that L is a constant independent of the policy and briefly expand the associated intuition.
>
> > The paper does not provide any theoretical performance or sample complexity advantage from using different maximum entropy objectives. All the theoretical results obtained in the paper are generally trivial.
>
> While formal performance or sample complexity bounds would be interesting, they are not standard requirements for introducing new RL objectives or algorithms. Our theoretical results are not intended as performance bounds. They relate our conditional objective to existing marginal entropy objectives and establish a contraction and fixed point structure that enables an off policy N step learning scheme. These results are, to the best of our knowledge, novel and necessary to derive the algorithm we propose. Empirically, our method typically converges faster and achieves better exploration metrics, which is already an indication of improved sample efficiency.
>
> > There are no assumptions in the formulations of the main theorems. Do I understand correctly that the theorems are always fulfilled for any spaces of states and actions?
>
> The theorems rely on two standard implicit assumptions. First, all KL divergences involved are finite. Second, the visitation distributions assign non zero probability to all state action pairs or features so that the entropies and KLs are well defined. These assumptions are standard in RL analyses.
>
> > The authors report that “all methods lead to similar control performance.” What is the practical impact in this case?
>
> The practical impact lies in exploration behavior, not only in final return. In many works, exploration is studied in reward free settings. Our experiments show that, while all methods reach similar returns, our conditional entropy objective yields better coverage within trajectories. This lets practitioners obtain comparable control performance while choosing an exploration pattern that is better aligned with the task.
>
> In light of these clarifications, we kindly ask the reviewer to reconsider their score and support acceptance of the paper.

---

### Meta-Review · Area_Chair_w71U · 2025-12-25

**Summary:**

My recommendation for the paper is 'Reject'. Overall, authors' rebuttal was helpful in clarifying misunderstanding about theoretical claims and results, but couldn't fully resolve concerns on the empirical evaluation of the paper. After the rebuttal, there still remain two major concerns around the empirical evaluation of the paper: (i) Limited evaluation only on Minigrid and (ii) Limited performance evaluation that leads to concern on the practical use-case of the proposed method. These concerns are not fully resolved after the rebuttal and this makes it difficult for me to recommend the acceptance of this paper. Moreover, there were several points made by Revieewer kSFD on the claims about the difference between on-policy and off-policy, and also about the difference between marginal and conditional entropy. It is encouraged that the authors incorporate the discussions on this into the future revision.

**Reviewer Concerns:**

Concerns addressed by the rebuttal
- Various clarifications on theoretical results & notations: Authors have done a great job on clarifying many misunderstandings of the reviewers
- Lack of exploration baselines: Authors did clarify that other exploration algorithms such as ICM or NGU can't be considered as baselines and I agree with that.

Outstanding concerns
- Evaluation only on Minigrid: All reviewers pointed out the experimental results are provided only on MiniGrid with discrete action spaces. While the authors have provided response on how to extend the current method to continuous action spaces, but the results are not provided in the rebuttal.
- Limited performance evaluation: Results also pointed out the empirical gain from the proposed idea is not clear, thus how the idea can be practically useful is not clear. While the authors have claimed and showed there is a notable increase of diversity in feature space, it is not clear how this can be actually useful. This perhaps can be further supported by fine-tuning performance on downstream tasks or other experimental designs, which were not done in rebuttal.
 - Formatting issue: Reviewer kSFD pointed out the issue with writing -- putting all figures in the appendix. While the authors claimed that this is a straightforward presentation fix but I don't fully agree with this. The quality of writing (clarity) is a major component in evaluating the quality of conference paper and this should have been already resolved in the rebuttal phase.

Unclear
- Discussion on (i) on-policy & off-policy and (ii) pros and cons of conditional entropy & marginal entropy: From the limited conversation between authors and reviewers, I don't think these discussions fully reached the consensus.

**Reviewer Scores:**

- Reviewer 2AbX (score 2): The reviewer's main concerns on the (i) MiniGrid-only results and (ii) practical use of the algorithm are not resolved in the rebuttal. I expect the reviewer would not have changed their score to acceptance.

- Reviewer kSFD (score 2): I don't think this reviewer's concerns are fully resolved -- (i) MiniGrid-only results, (ii) concern on the claim on on-policy & off-policy, and (iii) concern on using conditional entropy instead of marginal entropy. I expect the reviewer would not have changed their score to acceptance.

- Revieewer qoEK (score 2): I don't think one of the reviewer's main concerns on MiniGrid-only results is resolved. I expect the reviewer would not have changed their score to acceptance.

- Reviewer jJby (score 6): The reviewer was only the reviewer who originally gave the positive score. This reviewer also had concern on MiniGrid-only results, and based on this, I don't expect the reviewer would have fully championed the paper to be accepted.

---

### Decision · Program_Chairs · 2026-01-26

Reject